# Impacts of social distancing policies on mobility and COVID-19 case growth in the US

Gregory A. Wellenius[1,2,7], Swapnil Vispute[1,7], Valeria Espinosa[1,7], Alex Fabrikant[1,7], Thomas C. Tsai[3,4], Jonathan Hennessy[1], Andrew Dai [1], Brian Williams [1], Krishna Gadepalli[1], Adam Boulanger[1], Adam Pearce[1], Chaitanya Kamath[1], Arran Schlosberg[1], Catherine Bendebury[1], Chinmoy Mandayam[1], Charlotte Stanton[1], Shailesh Bavadekar[1], Christopher Pluntke[1], Damien Desfontaines[1,5], Benjamin H. Jacobson [4], Zan Armstrong [1], Bryant Gipson[1], Royce Wilson[1], Andrew Widdowson[1], Katherine Chou[1], Andrew Oplinger[1], Tomer Shekel[1], Ashish K. Jha[4,6] & Evgeniy Gabrilovich [1✉]

Social distancing remains an important strategy to combat the COVID-19 pandemic in the United States. However, the impacts of specific state-level policies on mobility and subsequent COVID-19 case trajectories have not been completely quantified. Using anonymized and aggregated mobility data from opted-in Google users, we found that state-level emergency declarations resulted in a 9.9% reduction in time spent away from places of residence. Implementation of one or more social distancing policies resulted in an additional 24.5% reduction in mobility the following week, and subsequent shelter-in-place mandates yielded an additional 29.0% reduction. Decreases in mobility were associated with substantial reductions in case growth two to four weeks later. For example, a 10% reduction in mobility was associated with a 17.5% reduction in case growth two weeks later. Given the continued reliance on social distancing policies to limit the spread of COVID-19, these results may be helpful to public health officials trying to balance infection control with the economic and social consequences of these policies.

[1] Google, Inc., Mountain View, CA, USA. [2] Department of Environmental Health, Boston University School of Public Health, Boston, MA, USA. [3] Department of Surgery, Brigham and Women's Hospital and Harvard Medical School, Boston, MA, USA. [4] Department of Health Policy and Management, Harvard T.H. Chan School of Public Health, Boston, MA, USA. [5] ETH Zurich, Zurich, Switzerland. [6] Brown University School of Public Health, Providence, RI, USA. [7]These authors contributed equally: Gregory A. Wellenius, Swapnil Vispute, Valeria Espinosa, Alex Fabrikant. ✉email: gabr@google.com

Social distancing remains a primary strategy for slowing the spread of the coronavirus disease 2019 (COVID-19) pandemic by reducing the frequency of close contact between individuals and thus minimizing the risk of transmission of the severe acute respiratory syndrome coronavirus 2. Prior experience with the 2009 H1N1 influenza and Ebola suggests that social distancing is effective in reducing disease transmission[1,2].

In China, officials engaged in an unprecedented quarantine of Hubei province to contain COVID-19 transmission out of the initial epicenter city of Wuhan[3–5]. As the pandemic spread to new clusters of infection in the United States, efforts at containment and then mitigation have been largely at the discretion of state and local governments, leading to a patchwork of directives to encourage social distancing. These policies have included state emergency declarations, work-from-home policies, school closures, closures of non-essential businesses and services, limits placed on large social gatherings, bans on in-restaurant dining, and shelter-in-place orders[6]. While multiple reports have examined the links between state-level social distancing policies, changes in mobility, and changes in case growth trajectories during the early phase of the pandemic[7–18], it remains unclear which state-level policies are most effective in mitigating the spread of the virus. Given the continued reliance on social distancing policies to limit the spread of COVID-19, systematically quantifying the impact of these policies may provide useful insights to government leaders trying to reduce the spread of the virus while minimizing the adverse economic and social impacts of restrictions. The recent availability of anonymized and aggregated mobility data provides an opportunity to quantify the effectiveness of individual policy interventions on both mobility and COVID-19 case growth[19].

In this work, we sought to: (1) quantify the effect on mobility of state emergency declarations, social distancing policies, and shelter-in-place orders, (2) identify which policies are most effective in reducing aggregate mobility, and (3) estimate the impact of changes in mobility on COVID-19 case growth in subsequent weeks. We found that state-level emergency declarations resulted in a 9.9% reduction in time spent away from places of residence, implementation of one or more social distancing policies resulted in an additional 24.5% reduction in mobility, and shelter-in-place mandates yielded an additional 29.0% reduction. Decreases in mobility were associated with substantial reductions in case growth 2–4 weeks later.

## Results

**Implementation of state-level policies.** Overall, we observed three waves of state-level responses to COVID-19: (1) a first wave occurring during the first 2 weeks of March, 2020 with state of emergency declarations, (2) a second wave during the week of March 16 where a variety of specific social distancing orders were enacted, and (3) a third wave during the last 2 weeks of March consisting of orders for residents to shelter in place (Fig. 1).

The first state of emergency related to COVID-19 was declared by Washington State on February 29, 2020 and the more recent ones by Oklahoma and Maine on March 15. Many states subsequently ordered that schools close (led by Louisiana and Virginia on March 13, 2020) and/or placed limits on specific activities and businesses in order to promote social distancing. Within a week, 48 states and Washington DC had implemented at least one social distancing policy. In 78% of states, the first social distancing order imposed was the closure of schools. On March 16, Nevada enacted orders advising residents to shelter in place, followed by California on March 19, 2020. By April 5, 80% of states had ordered residents to shelter in place.

**Impacts of social-distancing policies on population mobility.** Within each county, we applied a regression discontinuity analysis to estimate the impact of enactment of social distancing policies on mobility. To assess changes in population mobility, we used the same data that were used to prepare the Community Mobility Reports published by Google[20] (https://www.google.com/covid19/mobility). Mobility trends were aggregated to the county level (including Washington, DC and independent cities that are not otherwise included in county boundaries) and computed daily starting on January 3, 2020. The regression discontinuity approach provides a causal estimate of the short-term change in mobility in the week after versus before enactment of a specific policy. In the following sections, we report on the observed impacts on mobility of enactment of: (a) state emergency declarations, (b) different social distancing policies, and (c) shelter-in-place orders. We then evaluate the association between changes in mobility and subsequent changes in COVID-19 case growth rates. We chose the relative change in time spent away from places of residence as our primary mobility metric since that is on average proportional to the time at the risk of infection or contagion. We additionally considered the impacts of social distancing policies on relative changes in the number of visits to

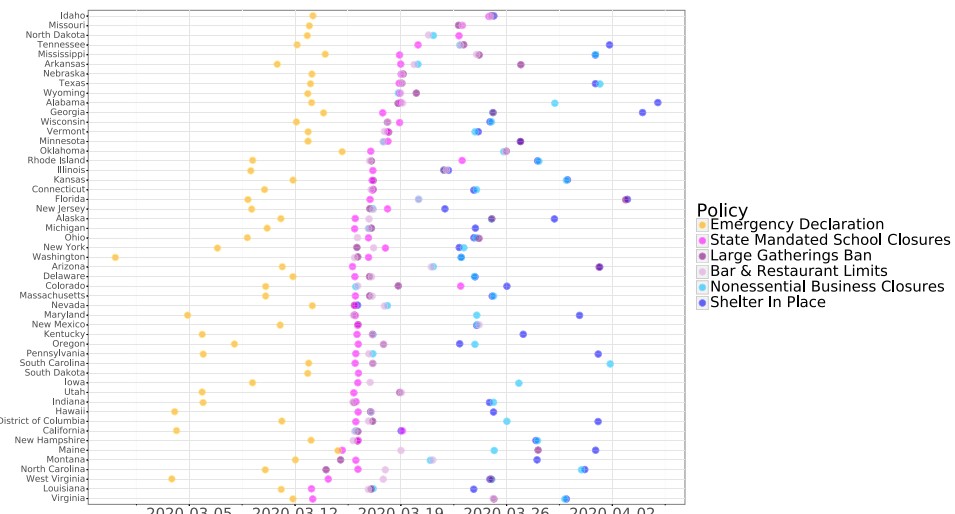

**Fig. 1 Implementation of state-level policies in response to COVID-19, ordered by date of first social distancing policy.** Although policies implemented through April 5, 2020 are shown, only effects of policies implemented through March 23, 2020 were evaluated.

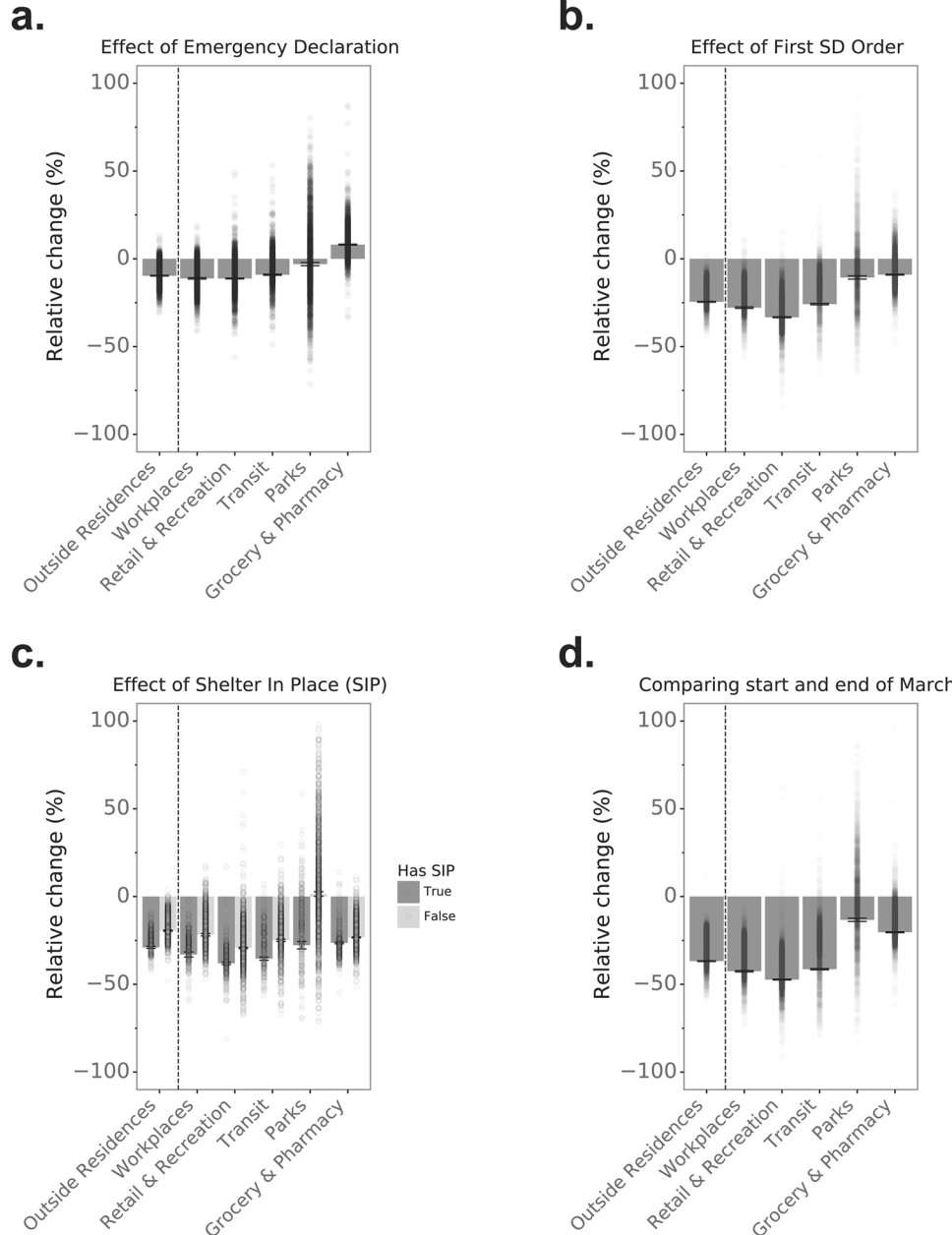

**Fig. 2 Impacts of state-level social distancing policies on population mobility. a** Average effect on mobility of state-wide emergency declaration orders. **b** Average effect on mobility of implementation of first state-wide social distancing (SD) policy in 49 states and Washington DC. Idaho is not included in this panel because its first social distancing order was enacted after March 23. **c** Average effect of shelter in place (SIP) order among 7 states that had issued such an order on or before March 23, 2020 and, for comparison, among the other states over a comparable time period (March 23–29 versus March 14–20, 2020). **d** Average change from the start of March (March 1–7, 2020) to the end of March (March 23–29, 2020). Each dot represents the estimated change in a given county (n = 2810) and each bar reflects the average change (and 95% confidence interval) across all counties.

places of work, grocery stores and pharmacies, retail and recreational sites, parks, and transit stations.

On average across the country, a declaration of a state of emergency was associated with a 9.9% (95% confidence interval [CI]: −10.1%, −9.7%) decrease in the time spent away from places of residence. State emergency declarations were also associated with 11.4% (95% CI: −11.8%, −11.0%) fewer visits to the workplace, 11.5% (−11.7%, −11.2%) fewer visits to retail and recreational sites, and 9.3% fewer (−9.6%, −9.0%) visits to transit stations in the following week (Fig. 2a). These changes in mobility are noteworthy given that emergency declarations did not

necessarily specifically call for increased social distancing and suggests that government messaging, news coverage, and/or actions observed in other countries could have influenced people's activities. Visits to parks were also affected by emergency declarations, with a small 3.5% (95% CI: −4.4%, −2.6%) average reduction. The smaller impact of emergency declarations on visits to parks versus other venues is likely at least partly explained by the seasonal transition to warmer weather during this period. On the other hand, emergency declarations coincided with a relative increase in visits to grocery stores and pharmacies of 8.2% (95% CI: 7.9%, 8.5%), consistent with news reports of individuals'

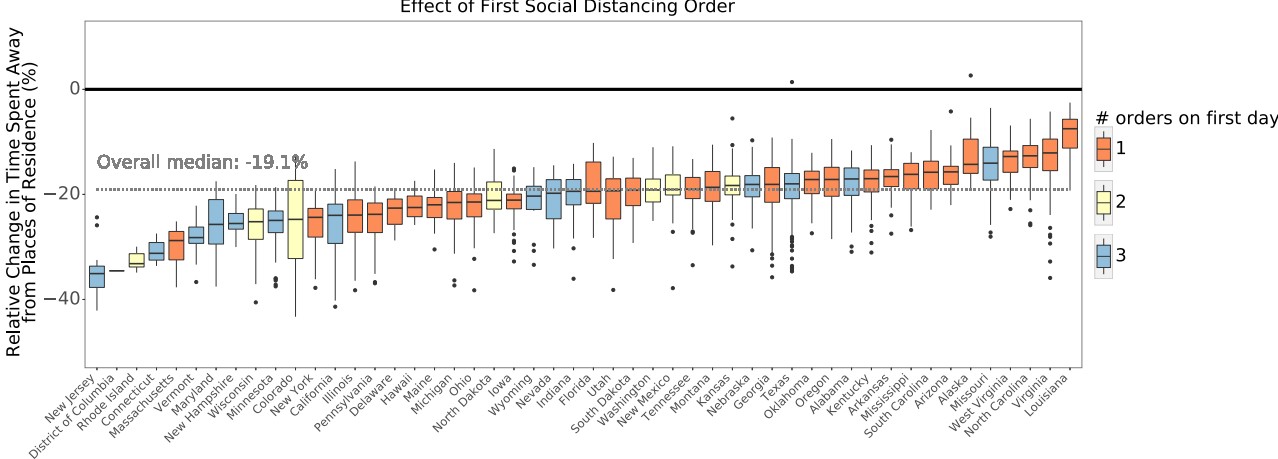

**Fig. 3 Effect of the first social distancing order on time spent away from places of residence by state.** Color coding reflects the number of social distancing policies that were simultaneously enacted in each state. Boxplots indicate the 25th and 75th percentiles (box extent) and the median (center line of each box) of county-specific changes. The whiskers extend from the hinge to the largest value no further than 1.5× interquartile range from the hinge. Dots represent outliers beyond the whiskers. N = 2810 counties in 50 US states and Washington, DC.

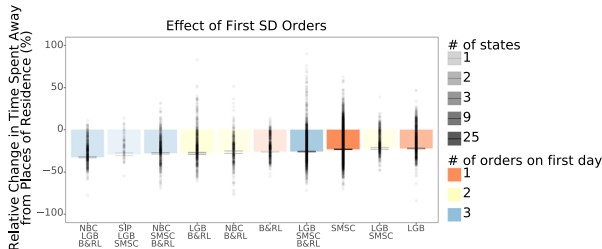

**Fig. 4 Association between different combinations of first social distancing orders on average time spent away from places of residence among 2810 counties in 50 US states and Washington, DC.** Note that only two combinations of measures are observed in more than three states. Each bar reflects the average change (and 95% confidence interval) as estimated from a multivariable regression model. LGB large gathering ban, SMSC state-mandated school closures, B&RL bar and restaurant limits, NBC nonessential business closures.

stocking up on dry goods, cleaning supplies, and medications at the end of February and early March in anticipation of further restrictions[21].

We next examined the impact on mobility of the first social distancing policies implemented in each state. We found that on average these orders resulted in additional reductions in mobility above and beyond the changes observed following the emergency declarations (Fig. 2b). Specifically, implementation of one or more social distancing policies resulted in a further 24.5% (95% CI: −24.7%, −24.3%) reduction in time spent away from places of residence, a further 33.0% (−33.3%, −32.8%) reduction in visits to retail and recreational outlets, and a further 27.9% (−28.3%, −27.5%) reduction in visits to work in the following week. The same pattern was evident for visits to parks, grocery stores, and pharmacies.

The impact of social distancing orders varied substantially between states (Fig. 3). For example, implementation of social distancing policies was associated with a 36% versus 12% decrease in the time spent away from places of residence in New Jersey versus Louisiana, although we note that differences in mobility between states may be due to a number of factors beyond social distancing policies. States that enacted multiple social distancing measures tended to experience greater reductions in mobility. The impact of social distancing orders also varied substantially across

counties within each state (Supplementary Table 1). This pattern of results was similar when considering other metrics of mobility (Supplementary Fig. 1). Our results are consistent with and extend findings of previous reports based on cell phone location data suggesting approximately a 50% reduction in individual mobility with variation across states[12].

Given that most states enacted multiple policies to encourage social distancing over a short time period, it is not possible to estimate the independent effects of individual policies. However, in secondary analyses we sought to identify the combinations of social distancing orders that were associated with greater changes in mobility (Fig. 4). Among states that initially implemented a single social distancing order, the most effective single order was imposing limits on bar and restaurant operations, which was associated with a 25.8% reduction (95% CI: −26.3%, −25.3%) in time spent away from places of residence. This result may suggest that restrictions on bars and restaurant operations discourage outings to other points of interests beyond restaurants, such as retail and recreation. The smallest reductions in time spent away from places of residence were observed in those states that only imposed state-mandated school closures and/or bans on large gatherings, suggesting that bar and restaurant limits and orders calling for the closure of non-essential businesses are critical to further reduce mobility.

We also considered the impact on mobility of state-wide orders to shelter in place. Among the 7 states that had issued shelter-in-place orders on or before March 23, we found substantial reductions in time spent away from places of residence and in visits to all categories of locations (Fig. 2c). Specifically, time spent away from places of residence was 29.0% (95% CI: −29.4%, −28.5%) lower in the week following implementation of shelter-in-place orders versus the prior week. Note that these changes are multiplicative over time as all states had already declared a state of emergency and implemented at least one social distancing policy. For comparison, we also show in Fig. 2c the change in mobility during the same time frame (March 23–29 versus March 14–20, 2020) among states that had and had not yet issued shelter-in-place orders by March 23. However, note that comparisons between states reflect the influence of a number of factors on mobility in addition to policy differences. The total average change in mobility comparing the end versus start of March (Fig. 2d) provides a measure of the cumulative impact of multiple interventions during this time.

**Population mobility and subsequent COVID-19 case growth**.
Finally, we quantified the association between changes in time spent away from places of residence and subsequent change in the growth rate of cases. Following the approach by Courtemanche et al.[7], we estimated case growth as the difference in the log of new cases from week to the next. We then fit a linear mixed effects regression model to estimate the impact of mobility changes on case growth at the county level, adjusting for temporal trends in case growth. We first examined the association between changes in mobility and subsequent changes in case growth 2 weeks later. We found that a 5% decrease in the time spent away from residences was associated with 9.2% fewer new cases of COVID-19 reported 2 weeks later (95% CI: −7.3%, −11.0%) (see Supplementary Table 4). A 10% decrease in mobility was associated with 17.5% fewer new COVID-19 cases reported 2 weeks later (95% CI: −14.1%, −20.9%). Changes in mobility were more strongly associated with changes in case growth rates 3 and 4 weeks later (Supplementary Table 4).

## Discussion

Overall, we find a strong relationship between the implementation of social distancing policy and decreasing mobility, which was in turn associated with decreased COVID-19 case growth. These estimates of timing and magnitude of the impacts of changes in mobility on case growth are consistent with previous reports[7–11,13–18]. For example, Gao et al.[8] showed that COVID-19 doubling time was associated with enactment of social distancing policies and correlated with home dwell time derived from SafeGraph mobility data. Similarly, Badr et al.[9] used anonymized cell phone tower data and found that reduced mobility was correlated with COVID-19 case growth rates in 25 US counties 2–3 weeks later. Our results add to this emerging body of evidence by providing quantitative estimates of the impacts of social distancing policies on multiple metrics of population mobility and subsequent changes in case growth in a very large population of Google users across the US.

These results should be interpreted in light of several important limitations. First, our mobility data are limited to smartphone users who have opted into Google's Location History feature. This data may not be representative of the changes in mobility of the population as a whole, and furthermore its representativeness may vary by location. Additionally, this data is only viewed through the lens of differential privacy algorithms, which were used to protect user anonymity when preparing the Community Mobility Reports[20]. Second, comparisons across rather than within locations are only descriptive since these regions can differ in substantial ways besides the policy environment. Third, our analyses are focused on state-level policies, but there is evidence of heterogeneous effects within states as some individual metropolitan areas and counties implemented specific social distancing policies prior to implementation of state-level policies. For example, Supplementary Fig. 4 shows that, in King County, WA, New York County, NY, and Santa Clara County, CA, implementation of the first county-level social distancing orders was accompanied by apparent reductions in time spent away from residences even before state-level social distancing orders were implemented. In Westchester County, NY, the first county and state-level social distancing orders coincided, but by then the town of New Rochelle, NY had already implemented some restrictions that may have led people across the county to voluntarily reduce their mobility. Thus, the impacts of state-level policies in areas where local or county social distancing orders were already in place may be smaller in magnitude than implied by our results.

In summary, using anonymized, aggregated, and differentially private data from Google users who opted into Location History, we found that state-mandated social distancing orders were effective in decreasing time spent away from places of residence, as well as reducing visits to work, grocery stores/pharmacies, and retail/recreational locations. While the majority of states declared states of emergency by early March, the emergency declaration per se had only a modest effect on mobility. In contrast, implementation of one or more specific social distancing orders was associated with an almost 25% additional reduction in time spent away from places of residence and a 33% additional reduction in visits to retail and recreational locations. These effects were evident in every state and in virtually every county. Although we were unable to comprehensively estimate the independent effects of different social distancing measures due to their close temporal proximity in each state, we found that those states that implemented multiple such measures experienced more pronounced declines in mobility. Furthermore, limits on bars and restaurants appeared to be the single most effective social distancing order to reduce mobility. Additionally, we replicated the findings of prior studies that changes in population-level mobility are strongly linked to changes in COVID-19 case growth in the subsequent weeks.

We conclude that state-based orders intended to promote social distancing appear to be highly effective in accomplishing the public health goals of encouraging individuals to minimize time away from their place of residence and thereby reduce the population risk of COVID-19 transmission. Our findings not only illustrate the effectiveness of specific social distancing orders but also quantify the magnitude of change in mobility and subsequent case growth that may result from such policies in the future. Moreover, our results highlight the potential utility of aggregate mobility data as a leading indicator of subsequent COVID-19 risk.

## Methods

Our analytic goals were to: (1) quantify the effect on mobility of state emergency declarations, social distancing policies, and shelter-in-place orders, (2) identify which policies are most effective in reducing aggregate mobility, and (3) estimate the impact of changes in mobility on COVID-19 case growth in subsequent weeks. We used a regression discontinuity approach to estimate the impact of state declarations of emergency and social distancing policies on mobility.

Data on state social distancing policies were obtained from official documents issued by state governors and health and education officials. Documents were linked from the Kaiser Family Foundation's State Data and Policy Actions Tracker and supplemented with manual searches of state public health websites. Dates of policy enactment were cross-checked with the American Enterprise Institute's COVID-19 Action Tracker and the New York Times Shelter in Place Tracker. Policies tracked were categorized as follows: (1) state-declared state of emergency, (2) state-mandated school closures, (3) state-mandated closing of non-essential businesses and services, (4) state-mandated limits on large gatherings, (5) state-imposed bans on in-restaurant service, and (6) state-imposed mandatory quarantines. State-mandated closing of non-essential businesses included any order closing gyms, theaters, and other businesses even if it did not extend to all non-essential businesses. Limits on large gatherings referred to any ban on gatherings larger than a certain number of people, though that threshold varied between states. For states that issued additional orders reducing the size of permitted gatherings, the date of the first such order was taken. Bans on in-restaurant service excluded mandatory reductions in restaurant capacity and included only those orders that prohibited any restaurant activity except pick-up and delivery. These bans often also included bars and clubs. Mandatory quarantine referred to any stay-at-home or shelter-in-place order that prohibited non-essential travel away from the home for all residents. Shelter-in-place orders specifically for high-risk individuals were excluded. Orders that went into effect at any time after 12:00 p.m. were considered to begin on the following day.

We obtained aggregated and anonymized data from Google users on mobile devices in all 50 states and Washington, DC who have opted into having their Location History data stored. The anonymized dataset[20] used for these analyses is the same as the one used to create the publicly available Google COVID-19 Community Mobility Reports (published at http://google.com/covid19/mobility on April 2, 2020). The Community Mobility Reports leverage signals such as relative frequency, time, and duration of visits to calculate metrics related to places of residence, work places, as well as several other categories of locations. The

anonymization process based on differential privacy was designed to ensure that no personal data, including an individual's location, movement, or contacts, can be derived from the resulting metrics.

Mobility data were aggregated to the county level (and Washington, DC) and available daily from January 3 through March 29, 2020. Changes for each day are compared to a pre-COVID baseline value for that same day of the week estimated over the period of 3 January 2020 through 6 February 2020. We chose to use the relative change in the average number of hours spent away from places of residence as the mobility metric of primary interest. This metric is estimated as 24 minus the population-averaged number of hours spent at places of residence and compared to the same baseline as used in the Community Mobility Reports. We considered relative changes in the number of visits to specific categories of points of interests as secondary metrics of mobility. Details of how these metrics are estimated have been published elsewhere[20].

Within each county, we applied a regression discontinuity analysis to estimate the relative change in time spent away from the place of residence (primary outcome) and the relative change in the number of visits to public locations (secondary outcomes) associated with: (1) declaration of a state of emergency, (2) ordering of one or more social distancing measures, and (3) orders for people to shelter in place. For each county, we compared the value of each metric in the week after the date of implementation versus the 7-day period 9–2 days prior to the date of implementation. We included a 2-day washout period prior to the implementation date given the public messaging that typically precedes implementation of these orders. Given that data until March 29 were used in these analyses, the effects of policies enacted on or before March 23 could be evaluated. Standard errors were calculated at the county level by first estimating the variance of the weekly average using February 1–28, 2020. The standard error of the relative change was calculated using the delta method. The state-level estimates reflect population-weighted aggregates of the county-specific estimates. The national estimates are a simple average of the state estimates.

We note that less populous counties are more likely to have days with missing data on visits to one or more categories of places (e.g., pharmacies), owing to the use of an anonymization procedure to protect user privacy. However, we believe that missing data has negligible effects on the state and national estimates provided because: (1) state-level estimates are weighted by county population, and populous counties are extremely unlikely to have any metrics that fall below the limits of detection, and (2) the (unweighted) correlations between the relative changes in the combined metrics used in this paper (e.g., grocery stores and pharmacies considered together), compared to an unbiased, but lower-coverage alternative (e.g., grocery stores only), are very high (>0.95).

Because each county and state is compared to its recent past, these estimates are causally interpretable. However, comparisons across counties or states are only descriptive since locations can differ substantially in terms of the proportion of the population that opted into Location History, the demographics of this group, the quality of the mobility data and of the Google Maps data about local establishments, and a number of other factors that may influence the observed changes in mobility beyond differences in the policy environment.

We obtained data on reported COVID-19 cases at the county level from the John Hopkins Coronavirus Resource Center. We then assessed the association between changes in mobility and subsequent changes in the number of new COVID-19 cases. Specifically, we fit a linear mixed effects model to estimate the change in the log of new cases from 1 week to the next in a county as a function of weekly mobility changes, the number of weeks elapsed since the county first reported its tenth infection (to account for the progression of the pandemic), and an indicator variable for the first week the county reported ten new cases (to mitigate potential noise resulting from changes in small case counts). We used a multilevel model where counties are nested within states since different states have differing policies that apply to all counties in a state. We not only hypothesized that case growth would be related to changes in population mobility 2 weeks earlier but also considered mobility changes 3 and 4 weeks earlier in separate models.

## Data availability

The anonymized and aggregated dataset analyzed herein was the same one that was used to create the publicly available Google COVID-19 Community Mobility Reports (first published at http://google.com/covid19/mobility on April 2, 2020). The data analyzed in this paper consisted of anonymized, aggregated, and differentially private counts of visits to places in different categories. The publicly available data reflects ratios computed using these counts. The information on dates of policy interventions was aggregated from publicly available data including from the Kaiser Family Foundation (https://www.kff.org/health-costs/issue-brief/state-data-and-policy-actions-to-address-coronavirus/ [accessed 2 April 2020]), the American Enterprise Institute (https://www.aei.org/covid-2019-action-tracker/ [accessed 2 April 2020]), and the New York Times (https://www.nytimes.com/interactive/2020/us/coronavirus-stay-at-home-order.html [accessed 2 April 2020]). Data on COVID-19 cases were obtained from the Johns Hopkins Coronavirus Resource Center (https://coronavirus.jhu.edu/). Data ethics: the usage of the data in this study complies with the terms of use of the Google COVID-19 Community Mobility Reports dataset. This project was determined to be not human subject research and was exempted by the institutional review board of the Harvard T.H. Chan School of Public Health and no ethical approval was required for the work presented by Google.

## Code availability

All analyses were performed using python 3.6.7 and graphics were created using the package plotnine 0.6.0.

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

## Acknowledgements

We are grateful to numerous colleagues at Google who helped launch the Community Mobility Reports. T.C.T. was supported by the William F. Milton Fund from the Harvard University Office of the Vice Provost for Research.

## Author contributions

G.A.W., S.V., A.B., A.P., C.S., S.B., K.C., A.O., T.S., and E.G. developed the concept. S.V., V.E., A.F., B.W., K.G., C.K., A.S., C.B., S.B., C.P., D.D., Z.A., C.M., B.G., R.W., and A.W.

computed the data. G.A.W., V.E., A.F., T.C.T., B.W., K.G., A.B., A.P., A.S., C.B., B.J., Z.A., A.O., T.S., A.K.J., and E.G. designed the experiments. G.A.W., V.E., A.F., T.C.T., J.H., A.D., B.W., C.B., and Z.A. analyzed the data and performed statistical analysis. G.A.W., S.V., V.E., A.F., T.C.T., J.H., A.D., B.W., K.G., A.B., A.P., C.K., A.S., C.B., C.M., S.B., C.P., D.D., Z.A., B.G., R.W., A.W., A.O., T.S., and E.G. accessed the data. All authors contributed to writing the manuscript. G.A.W., S.V., V.E., and A.F. contributed equally to this manuscript and therefore are listed as co-first authors.

## Competing interests

The authors declare no competing interests.
