## [Peer Review File · Nature Communications]

Reviewer comments first round –

Reviewer #1 (Remarks to the Author):

In this manuscript, the authors analyzed aggregated data from Google users across the U.S. who have opted to share Location History. Using these data and information on the dates of implementation of various social distancing policies, the authors i) quantify the average effect on mobility of social distancing policies and ii) gain insights into which policies are most effective.

The article is organized well, and easy to read. The topic and analysis are relevant and interesting, the authors definitely have previous experience working with similar topics.

In addition, the data is technically sound and all the analysis made provide strong evidence for its conclusions, with appropriate statistical analyses and discussions regarding the limitations of the approach implemented. The results are novel and important to scientists and policymakers worldwide working to minimize the risk of COVID-19 in the future months. For these reasons, I recommend the publication of the manuscript.

Reviewer #2 (Remarks to the Author):

The manuscript provides an interesting dataset in US during the Covid-19 epidemic. The analysis of the data is interesting and well-done.

However, the novelty of the data/approach is not as high as claimed. Similar analysis are been performed by other authors

<https://www.networkscienceinstitute.org/publications/reshaping-a-nation-mobility-commuting-and-contact-patterns-during-the-covid-19-outbreak>

https://www.medrxiv.org/content/10.1101/2020.05.07.20092353v1#disqus_thread

The present analysis is of good quality and merits to be published in a good journal, maybe, after improvements. But it lacks the relevance and novelty expected for Nature Communications articles.

Reviewer #3 (Remarks to the Author):

Review: Impacts of US State-Level Social Distancing Policies on Aggregated Mobility Metrics During the COVID-19 Pandemic – Nature Comm

This paper uses Google's aggregated mobility data to study the impact of state-level policies on different kinds of mobility behaviors and quantifies specific reductions for those behaviors. It is nice to see the mobility research work from a team of Google staff. I also appreciate that Google has made the Community Mobility Reports data set available for COVID-19 research.

My major concern is that although the authors have assessed the impact of policies on mobility, they don't link mobility to some more substantial issues such as COVID-19 cases, economic or social consequences. I hope the authors would be able to extend their analysis on those significant issues (or even one of those issues). I do feel that such additional analyses are of critical importance though more work is needed. It would be nice to see different effects of mobility behaviors on COVID-19 spread or economic consequences. Some recent relevant papers are provided in the references below.

The authors don't explain the details of their mobility data clearly. What is the definition of "Outside Residences"? What is the relationship between this and the other five mobility measures

(i.e., workplaces, retail & recreation, transit, parks, grocery & pharmacy). I also think the three types of mobility, retail & recreation, transit, parks, grocery & pharmacy, are not well classified. For instance, retail stores should include grocery stores, and recreation sites might be grouped together with parks.

The Authors mentioned that "To our knowledge there have been no systematic analyses comparing the impacts of different policies on mobility, or providing quantitative estimates of the magnitude of change in mobility associated with such policies." In fact, several papers studied such issues in different countries in the references below.

What is the total effect of emergency declaration orders, social distancing (SD) policy, and shelter in place (SIP) order together on mobility reduction?

The regression discontinuity model used in this paper should be specified in Online Methods section. This might be a methodological advantage, compared with other existing papers. Have you considered weekly effects in modeling mobility behaviors?

Can you provide specific data evidence regarding your argument about "limits on bars and restaurants appeared to be the single most effective social distancing order"? (The last sentence of the second last paragraph)

References:

Bonaccorsi, Giovanni et al., "Economic and Social Consequences of Human Mobility Restrictions under COVID-19," Working Paper (https://papers.ssrn.com/sol3/papers.cfm?abstract_id=3573609, 2020)

Chen, M. Keith et al., "Causal Estimation of Stay-at-Home Orders on SARS-CoV-2 Transmission," Working Paper (<https://arxiv.org/abs/2005.05469>, 2020)

Fang, H., L. Wang, Y. Yang, "Human Mobility Restrictions and the Spread of the Novel Coronavirus (2019-nCoV) in China " Working Paper No. 26906 (National Bureau of Economic Research, 2020).

Fowler, J. H., S. J. Hill, R. Levin, N. Obradovich, "The Effect of Stay-At-Home Orders on COVID-19 Infections in the United States," Working Paper (<https://arxiv.org/abs/2004.06098>, 2020).

Holtza, David et al., "Interdependence and the Cost of Uncoordinated Responses to COVID-19," Working Paper

(http://ide.mit.edu/sites/default/files/publications/Interdependence_COVID_522.pdf, 2020)

Jia, J.S. et al., "Population Flow Drives Spatio-Temporal Distribution of COVID-19 in China," Nature, 2020 (<https://www.nature.com/articles/s41586-020-2284-y>)

Point by Point Response to Reviewers

Reviewer #1 (Remarks to the Author):

Comment: *In this manuscript, the authors analyzed aggregated data from Google users across the U.S. who have opted to share Location History. Using these data and information on the dates of implementation of various social distancing policies, the authors i) quantify the average effect on mobility of social distancing policies and ii) gain insights into which policies are most effective.*

The article is organized well, and easy to read. The topic and analysis are relevant and interesting, the authors definitely have previous experience working with similar topics.

In addition, the data is technically sound and all the analysis made provide strong evidence for its conclusions, with appropriate statistical analyses and discussions regarding the limitations of the approach implemented. The results are novel and important to scientists and policymakers worldwide working to minimize the risk of COVID-19 in the future months. For these reasons, I recommend the publication of the manuscript.

Response: Thank you for the positive feedback on our manuscript.

Reviewer #2 (Remarks to the Author):

Comment: *The manuscript provides an interesting dataset in US during the Covid-19 epidemic. The analysis of the data is interesting and well-done.*

Response: Thank you for the positive feedback.

Comment: *However, the novelty of the data/approach is not as high as claimed. Similar analysis are been performed by other authors*

<https://www.networkscienceinstitute.org/publications/reshaping-a-nation-mobility-commuting-and-contact-patterns-during-the-covid-19-outbreak>

https://www.medrxiv.org/content/10.1101/2020.05.07.20092353v1#disqus_thread

Response: Thank you for the suggestion. We now reference an expanded set of relevant reports, including those suggested by the reviewer and highlight more precisely what our study adds. The list of newly added references appears below at the end of this response letter.

Comment: *The present analysis is of good quality and merits to be published in a good journal, maybe, after improvements. But it lacks the relevance and novelty expected for Nature Communications articles.*

Response: We thank the reviewer for the feedback. However, we feel that our findings are both relevant and novel. From the standpoint of relevance, the US has entered a phase of the COVID-19 pandemic with record-breaking numbers of new COVID-19 cases and rising hospitalization and mortality rates. Now, more than ever, establishing a clear relationship between social distancing policy and changes in mobility as well as rates of COVID-19 case growth highlights the effectiveness of a key public health strategy. This is particularly salient as states and municipalities are considering whether to re-institute social distancing measures. Our findings therefore can have immediate impact for policy makers.

While the literature linking changes in mobility and COVID-19 cases is growing, the novelty of our approach is twofold: 1) linking policy to mobility to COVID-19 cases and 2) establishing specific patterns of mobility response. While not causal, our approach does help assess potential mechanisms of causality. For example, by establishing that restaurant bans were associated with the largest decrease in mobility as an isolated policy, we have shown the potential for more targeted social distancing policy than wide-scale lockdowns. This adds new empirical evidence to recent work suggesting targeted policies are superior to holistic closures, and it has had immediate policy-relevance. The Florida Department of Public Health has cited our preprint work to inform decisions to re-institute restaurant bans during the summer of 2020. While prior studies have been performed using proprietary datasets of aggregated mobility, our study uses a public dataset with strict differential privacy safeguards. This research demonstrates a potential ongoing use case for Google's mobility data to further public health and public policy efforts.

Reviewer #3 (Remarks to the Author):

Comment: *This paper uses Google's aggregated mobility data to study the impact of state-level policies on different kinds of mobility behaviors and quantifies specific reductions for those behaviors. It is nice to see the mobility research work from a team of Google staff. I also appreciate that Google has made the Community Mobility Reports data set available for COVID-19 research.*

Response: thank you for the positive feedback.

Comment: *My major concern is that although the authors have assessed the impact of policies on mobility, they don't link mobility to some more substantial issues such as COVID-19 cases, economic or social consequences. I hope the authors would be able to extend their analysis on those significant issues (or even one of those issues). I do feel that such additional analyses are of critical importance though more work is needed. It would be nice to see different effects of mobility behaviors on COVID-19 spread or economic consequences. Some recent relevant papers are provided in the references below.*

Response: Thank you for this suggestion. As you suggested, in the revised manuscript we further study the connection between the changes in mobility and the subsequent changes in

COVID-19 case growth. We found that a 5% decrease in the time spent away from residences was associated with 9.2% fewer new cases of COVID-19 reported 2 weeks later (95% CI: -7.3%, -11.0%). A 10% decrease in time spent away from places of residence was associated with 17.5% fewer new COVID-19 cases reported 2 weeks later (95% CI: -14.1%, -20.9%). These estimates of timing and/or magnitude of the impacts of changes in mobility on case growth are consistent with peer-reviewed and preliminary reports published elsewhere.

Comment: *The authors don't explain the details of their mobility data clearly. What is the definition of "Outside Residences"? What is the relationship between this and the other five mobility measures (i.e., workplaces, retail & recreation, transit, parks, grocery & pharmacy). I also think the three types of mobility, retail & recreation, transit, parks, grocery & pharmacy, are not well classified. For instance, retail stores should include grocery stores, and recreation sites might be grouped together with parks.*

Response: Thank you for the opportunity to clarify important details about the mobility data. The anonymized dataset used for these analyses is the same as the one used to create the publicly-available Google COVID-19 Community Mobility Reports. The Community Mobility Reports leverage signals such as relative frequency, time and duration of visits to calculate metrics related to places of residence, places of work and public locations visited by Location History users. Changes for each day are compared to a pre-COVID baseline value for that same day of the week in the period of 2020-01-03 through 2020-02-06. For example, a value of -2 on a given Monday indicates that there were an estimated 2% fewer visits compared to the median number of visits to those same locations on the Mondays of the reference period. We chose to use relative change in the average number of hours spent away from places of residence as the mobility metric of primary interest. This metric is estimated as 24 minus the population-averaged number of hours spent at places of residence and compared to the same baseline as used in the Community Mobility Reports. We considered relative changes in the number of visits to specific categories of points of interests as secondary metrics of mobility. Details of how these metrics are estimated have been published elsewhere (20, 21 in main text). In the revised manuscript, we have added additional details about the mobility metrics to the Online Methods.

Comment: *The Authors mentioned that "To our knowledge there have been no systematic analyses comparing the impacts of different policies on mobility, or providing quantitative estimates of the magnitude of change in mobility associated with such policies." In fact, several papers studied such issues in different countries in the references below.*

Response: Thanks for this comment. In the revised manuscript we more explicitly place these analyses in the context of prior work. The second paragraph now contains the following: **"While multiple reports have examined the links between state-level social distancing policies, changes in mobility, and changes in case growth trajectories during the early phase of the pandemic (7-19), it remains unclear which state-level policies are most effective in mitigating the spread of the virus. Given the continued reliance on social distancing**

policies to limit the spread of COVID-19, systematically quantifying the impact of these policies may provide useful insights to government leaders trying to reduce the spread of the virus while minimizing the adverse economic and social impacts of restrictions.”

***Comment:** What is the total effect of emergency declaration orders, social distancing (SD) policy, and shelter in place (SIP) order together on mobility reduction?*

Response: Thanks for the comment. We have added Figure 2D to show the cumulative impact of multiple social distancing measures on mobility.

***Comment:** The regression discontinuity model used in this paper should be specified in Online Methods section. This might be a methodological advantage, compared with other existing papers. Have you considered weekly effects in modeling mobility behaviors?*

Response: Thank you for this suggestion. We have expanded the Online Methods section to further explain our approach.

***Comment:** Can you provide specific data evidence regarding your argument about “limits on bars and restaurants appeared to be the single most effective social distancing order”? (The last sentence of the second last paragraph)*

Response: This statement is based on the results shown in Figure 4 that among states that implemented only a single social distancing order, the implementation of limits on bars and restaurants was associated with the greatest relative decrease in mobility.

References added in the manuscript:

11. Bonaccorsi, Giovanni et al., “Economic and Social Consequences of Human Mobility Restrictions under COVID-19,” (PNAS, 2020)
12. Klein, B. et al. Reshaping a nation: Mobility, commuting, and contact patterns during the COVID-19 outbreak. (Network Science Institute, 2020).
13. Chen, M. Keith et al., “Causal Estimation of Stay-at-Home Orders on SARS-CoV-2 Transmission,” Working Paper (<https://arxiv.org/abs/2005.05469>, 2020)
14. Fang, H., L. Wang, Y. Yang, "Human Mobility Restrictions and the Spread of the Novel Coronavirus (2019-nCoV) in China " (Journal of Public Economics, 2020).
15. Fowler, J. H., S. J. Hill, R. Levin, N. Obradovich, "The Effect of Stay-At-Home Orders on COVID-19 Infections in the United States," Working Paper (<https://arxiv.org/abs/2004.06098>, 2020).
16. Holtz, David et al., “Interdependence and the Cost of Uncoordinated Responses to COVID-19,” (PNAS, 2020)
17. Jia, J.S. et al., “Population Flow Drives Spatio-Temporal Distribution of COVID-19 in China,” Nature, (2020).
18. Chang, S. et al. Mobility network models of COVID-19 explain inequities and inform reopening. Nature, (2020).

Reviewer comments, second round:

Reviewer #1 (Remarks to the Author):

The authors have updated the manuscript considering the points raised in the evaluation of the paper. I recommend the publication of this new version in Nature Communications.

Reviewer #3 (Remarks to the Author):

The authors have addressed my review comments and made a significant revision. I would recommend the manuscript for publication.